# Rethinking HOI Evaluation for both Vision–Language Models and HOI-specific Methods

## Abstract

Human-object interaction (HOI) detection has traditionally been approached with task-specific models, sometimes augmented by early vision-language models (VLMs) such as CLIP. With the rise of large, generative VLMs, however, a natural question emerges: can standalone VLMs effectively perform HOI detection, and how do they compare to specialized HOI methods? Addressing this requires a benchmarking dataset and protocol that support both paradigms. Existing benchmarks such as HICO-DET were developed before modern VLMs and rely on exact label matching. This clashes with generative outputs, which may yield multiple equally valid interpretations. For example, in a single image, a person mid-motion with a frisbee might plausibly be described as "throwing" or "catching," yet only one is annotated as correct. Such rigid evaluation penalizes valid predictions from both VLMs and HOI-specific methods, but disproportionately underestimates VLM performance because their outputs are less constrained. We introduce a new benchmarking dataset that reformulates HOI detection as a multiple-answer multiple-choice task. It emphasizes challenging scenarios by (i) including a higher proportion of multi-person scenes where individuals perform different interactions, (ii) removing overly simple cases, and (iii) curating hard negative choices. This makes the benchmark more challenging than prior HOI datasets, while still supporting systematic evaluation of both standalone VLMs and HOI-specific under a unified protocol. Our results show that large VLMs already surpass state-of-the-art HOI-specific methods across most metrics, while analysis further uncovers key limitations: VLMs often misattribute surrounding people's interactions to the target person and struggle in complex multi-person or occluded scenarios.

## 1 Introduction

Human-object interaction (HOI) detection is a long-standing problem in computer vision, traditionally addressed with task-specific models. Recent works have incorporated early vision–language models (VLMs) such as CLIP Radford et al. (2021) and BLIP Li et al. (2022a) as encoders to provide aligned image–text features within HOI frameworks Liao et al. (2022); Ning et al. (2023); Mao et al. (2024); Cao et al. (2024); Yuan et al. (2023). In parallel, the rapid progress of large, generative VLMs (e.g., Qwen2.5-VL Bai et al. (2025), InternVL3 Zhu et al. (2025)) has demonstrated that they can directly interpret complex visual scenes, often producing fluent descriptions that include human–object interactions Bai et al. (2025); Shahriar et al. (2024); Chen et al. (2024); Yang et al. (2023). This contrast between HOI-specific pipelines and general-purpose standalone VLMs highlights the need to reconsider how HOI detection is evaluated across these two paradigms.

Unlike earlier approaches that use VLMs merely as encoders within HOI-specific pipelines, in this paper we focus on VLMs as standalone models (unless otherwise noted). This distinction raises a central question: can general-purpose VLMs, applied directly in an end-to-end manner, perform HOI detection on par with or even better than specialized HOI methods Lei et al. (2025); Li et al. (2024a); Kim et al. (2025)? The answer is crucial for understanding the true state of progress in HOI detection and for clarifying how future research should position HOI-specific methods relative to

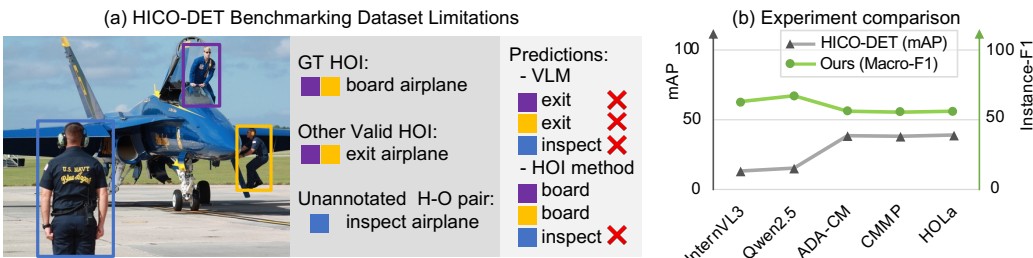

Figure 1: (a) Existing benchmarks require exact matches with ground-truth annotations (e.g., "board airplane"), but these annotations are often incomplete, missing plausible interactions such as "exit airplane" or omitting human-object pairs altogether (e.g., the person in the blue box and the airplane). (b) Comparison of state-of-the-art VLMs (InternVL3 Zhu et al. (2025), Qwen2.5-VL-32B Bai et al. (2025)) and HOI-specific methods (ADA-CM Lei et al. (2023), CMMP Lei et al. (2024c), HOLa Lei et al. (2025)). Results are shown using Instance-F1 in our benchmark ( *Setting 3*) versus mean Average Precision (mAP) in the HICO-DET benchmark, highlighting the performance gap caused by current evaluation protocols.

general-purpose VLMs. Addressing this question requires a new benchmark that can fairly evaluate both paradigms under a unified protocol.

At first glance, existing HOI benchmarks such as HICO-DET Chao et al. (2018) may appear sufficient for comparing VLMs and HOI-specific methods, since both can ultimately output bounding boxes and HOI classes. In practice, however, these benchmarks were designed before modern VLMs and enforce strict evaluation criteria that require exact matches with annotated HOI classes, rejecting any alternative interpretation. Such evaluation criteria align with HOI-specific models, which operate on a fixed label space, but they are poorly suited to VLMs, whose generative outputs can naturally produce multiple equally valid descriptions. For example, as shown in Fig. 1(a), a person mid-motion with an airplane might reasonably be described as "boarding" or "exiting." Yet when only one label is annotated, the other, though equally valid, is counted as an error. This incomplete annotation practice penalizes correct predictions and systematically underestimates model capability, especially for VLMs.

A further limitation lies in the sparsity of HOI annotations, which becomes especially severe in multi-person and multi-object scenes. Because HOI tasks involve combinations of actions and objects, the annotation space grows exponentially, making comprehensive labeling infeasible Hou et al. (2020; 2021); Lei et al. (2024a;c). As illustrated in Fig. 1(a), the person in the blue box interacting with the airplane is not annotated in the ground truth, so any valid prediction involving this pair is incorrectly penalized as an error. This problem affects both VLMs and HOI-specific methods, since all models are judged against incomplete ground truths, leading to underestimated performance. Fig. 1(b) highlights the impact: HOI-specific methods remain below 50% mAP, while general-purpose VLMs drop to around 15%. This striking gap underscores the inadequacy of current HOI benchmarks and the need for new protocols that can fairly evaluate both paradigms.

To overcome these limitations, we introduce a new benchmark that reformulates HOI detection as a multiple-answer, multiple-choice task. Each question includes annotated ground-truth interactions as positive choices and a curated set of negatives. We begin with a multi-stage filtering process using several state-of-the-art VLMs to eliminate plausible false negatives. To further enhance the difficulty, we refine the dataset by excluding overly simple scenarios, such as a single person in a clean background performing an unambiguous action or multiple people engaging in the same interaction without ambiguity. From the remaining images, we construct hard negatives, drawing on interactions performed by surrounding people and fine-grained visually similar actions, thereby making the benchmark challenging.

While the multiple-choice reformulation with curated negatives helps address the issue of missing interaction labels, the sparsity of HOI annotations remains a challenge. To mitigate this, our benchmark introduces three complementary evaluation settings. In some settings, evaluation is restricted to the target person, ensuring that unannotated interactions involving other individuals are excluded. In others, the scope is broadened to multi-person interaction recognition and detection, capturing

more complex scenarios. Together, these settings provide a more comprehensive assessment of HOI understanding across different levels of difficulty.

Using our benchmarking dataset, we systematically evaluate both standalone VLMs and HOI-specific methods under a unified protocol, enabling direct comparison between the two paradigms. Our experiments reveal that large VLMs, even in zero-shot evaluation, already surpass state-of-the-art HOI-specific methods across most metrics (e.g., achieving up to +16.65% Macro-F1) and remain competitive in the rest. Smaller VLMs, however, perform only on par with HOI-specific models and exhibit sharp drops when detection ability is tested. Furthermore, our analysis highlights persistent limitations of VLMs: they frequently misattribute interactions of surrounding people to the target person and struggle with detection in complex scenarios involving multiple individuals or occlusions.

In summary, our work makes the following contributions:

- We introduce the first benchmarking dataset that jointly evaluates standalone general-purpose VLMs and HOI-specific methods. By reformulating HOI detection as a multiple-answer, multiple-choice task with curated negatives, our dataset resolves the ambiguity and annotation sparsity that limit existing benchmarks such as HICO-DET. This provides a foundation for future research on both VLM-based and HOI-specific approaches.

- We conduct a comprehensive benchmarking study of state-of-the-art VLMs and HOI-specific methods on our dataset, providing the first direct comparison under a unified protocol. This evaluation clarifies how far large VLMs have advanced relative to specialized HOI models, while also highlighting the relative strengths and remaining gaps between the two paradigms.

- Through our analysis, we present several key findings: large VLMs already establish new state-of-the-art performance on HOI detection without task-specific training; their errors often arise from confusing the target person's interactions with those of nearby people; and their detection ability degrades in complex multi-person or occluded scenarios. These findings identify concrete directions for advancing HOI detection, especially regarding the role of VLMs.

## 2 NEW BENCHMARK

### 2.1 PRELIMINARY

We construct our benchmark on top of HICO-DET Chao et al. (2018), one of the most widely used datasets for HOI detection Tamura et al. (2021); Wu et al. (2023); Hou et al. (2020); Yuan et al. (2022); Tu et al. (2023). HICO-DET consists of 47,774 images, split into 38,118 for training and 9,658 for testing, spanning 600 HOI classes. The dataset is long-tailed, with many rare classes Hou et al. (2021); Wang et al. (2022a); Yang et al. (2024), and its training and testing distributions are almost identical. We quantify this overlap by computing the KL divergence between the two splits, which is only 0.088. Such similarity risks encouraging models to rely on dataset priors rather than true visual understanding, a phenomenon also observed in related tasks Agrawal et al. (2018).

In addition, HICO-DET presents two further challenges. First, 68 of the 600 HOI classes are inherently ambiguous in static images because temporal dynamics are absent (e.g., distinguishing between "boarding" and "exiting" in Fig. 1). A detailed discussion and the full class list are provided in Appendix A. Second, 4,800 out of the 9,658 test images contain multiple people, as identified using a VLM Bai et al. (2025). In such cases, exhaustive HOI annotation is infeasible due to the quadratic growth of possible human–object pairs Hou et al. (2020); Lei et al. (2024c). This leads to sparse labeling, a phenomenon also reported in previous multi-person vision tasks Gupta et al. (2019a); Niitani et al. (2019); Zhang et al. (2019); Suri et al. (2023). These issues make evaluation protocols that require exact matches to annotated HOI classes problematic, since the annotations are often ambiguous or incomplete.

Table 1 compares the distribution of single-person and multi-person cases across existing HOI benchmarks. Our dataset contains a substantially larger proportion of multi-person scenes involving different HOIs (38.9%), supporting our claim that the selected test cases are more challenging.

| Dataset | Singlel-Person Single Obj ↓ | Multi-Person Diff. HOI ↑ | Applicable to HOI methods | Applicable to VLMs | Multi-Class HOI Prediction |
|---|---|---|---|---|---|
| HICO-DET | 60.2% | 7.5% | ✓ | ✗ | ✓ |
| V-COCO | 51.2% | 22.5% | ✓ | ✗ | ✓ |
| SWiG-HOI | 62.2% | 0.0% | ✓ | ✗ | ✓ |
| Bongard-HOI | 100.0% | 0.0% | ✓ | ✓ | ✗ |
| Ours | **33.3%** | **38.9%** | ✓ | ✓ | ✓ |

Table 1: Comparison between existing HOI benchmarks and ours.

At the same time, simpler cases are reduced. For instance, the proportion of single-person single-object HOIs decreases from 60.2% in the HICO-DET test set to 33.3% in ours. Within the multi-person subset, same-HOI scenes account for 76.0% of the HICO-DET test set but only 28.8% in our benchmark, highlighting the increased diversity of interactions. Unlike HICO-DET, V-COCO, and SWiG-HOI, our benchmark is explicitly designed to support both HOI-specific methods and VLMs, providing the first evaluation setting that enables direct comparison across the two paradigms.

## 2.2 DATASET CONSTRUCTION

**Task Reformulation**   To address the limitations of exact-match evaluation, we reformulate HOI detection as a multiple-answer, multiple-choice task. For each human-object pair in an image, we construct a question with four candidate options, and the model must identify all correct ones. Since a person may simultaneously engage in multiple valid interactions (e.g., *hold knife* and *cut with knife*), a question can include more than one positive answer. In our benchmark, positive choices are taken from ground-truth annotations in HICO-DET, while negative choices are curated to exclude plausible but unannotated interactions. This reformulation reduces the likelihood of penalizing valid predictions and provides a consistent basis for evaluating both VLMs and HOI-specific methods.

**Negative Choice Construction**   For each human-object pair in an image, we first build a candidate pool by collecting all plausible actions for the object and removing its ground-truth labels. To reduce false negatives caused by incomplete annotations in HICO-DET, we apply a multi-stage VLM-based pipeline. Because a single VLM often struggles with negative choice judgments, we combine multiple VLMs in sequence to improve robustness and quality. GPT-4.1 OpenAI (2023) initially separates candidates into semantically consistent or inconsistent with the image, and we retain only the inconsistent ones. The retained pool is then refined using Qwen2.5-VL-32B Bai et al. (2025). Since Qwen2.5-VL is also one of our evaluation baselines, directly adopting its judgments could bias the benchmark. To avoid this, negatives rejected by Qwen2.5-VL are cross-checked with GPT-4o Hurst et al. (2024), ensuring that semantically plausible but challenging cases are preserved rather than mistakenly discarded. Each question follows a fixed four-choice format with randomized option order, allowing multiple correct answers. Randomization prevents positional bias, as large language models are known to be sensitive to option ordering Pezeshkpour & Hruschka (2024); Zheng et al. (2024).

**Hard Case Refinement**   Although the multi-stage filtering ensures valid negatives, it often removes difficult cases, leaving the benchmark clean but not sufficiently challenging. To address this, we introduce a manual refinement stage, performing manual checks on every question (i.e., every image in the benchmark). We exclude visually unambiguous and simple cases, such as images with only one person and one salient object against a clean background where the interaction is obvious at a glance (e.g., riding a motorcycle), and scenes in which multiple people clearly perform the same interaction. From the remaining images, we strengthen the benchmark in two ways. First, we include harder positives by allowing multiple temporally plausible actions to be correct (e.g., both *boarding* and *exiting* like in Fig. 1). Second, we add hard negatives, including interactions of surrounding people that differ from those of the target person, and fine-grained distinctions between visually similar actions, such as *holding a person* versus *hugging a person*. This refinement increases the difficulty of the dataset while maintaining the validity of negative choices, ensuring that the benchmark captures the complexity of real-world HOI understanding.

**Test Set Redistribution**   Our benchmark is constructed from the HICO-DET test set, which is widely used for evaluating HOI detection methods and ensures compatibility with models trained on HICO-DET. However, as shown in Sec. 2.1, the training and test distributions of HICO-DET are highly similar, which risks inflating performance by allowing HOI-specific models to exploit dataset

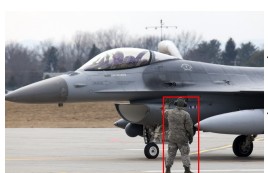

Setting 1:
Select interactions for all persons in the image from choices [GT: A,B,C,D]
_______________________________________________________
Setting 2:
Select interactions for the target person (box given) from choices [GT: A]
_______________________________________________________
Setting 3:
First detect all persons, then Select interactions for the target person from choices [GT: A]
Choices: (A) inspect airplane; (B) ride airplane; (C) fly airplane; (D) sit on airplane

Figure 2: Example questions in our new HOI benchmark, illustrated under three evaluation settings.

priors rather than demonstrating genuine HOI understanding Agrawal et al. (2018). To mitigate this issue, we redistribute the test set to create a distribution that is less skewed and more distinct from the training set. The KL divergence between the HICO-DET training and test splits is only 0.088, whereas the divergence between the training split and our redistributed test set increases to 0.629. Moreover, when head and tail classes are defined using HICO-DET training frequencies, the ratio between the top-20 head and bottom-20 tail classes drops from 80.2 (HICO-DET test set) to 7.1 in ours, indicating a substantially reduced long-tail bias. Although this redistribution results in a smaller test set, it provides a more valid evaluation by discouraging HOI-specific methods from over-relying on training priors.

**Dataset Summary** Our benchmark contains 1,274 images selected from the HICO-DET test set. The size is smaller than the original HICO-DET test set due to the removal of overly simple cases and the redistribution process, where many rare classes contribute only a few samples. We do not expand the dataset by merging multiple HOI benchmarks, since our goal is to support both VLMs and HOI-specific methods, and the latter cannot be consistently evaluated across datasets with different predefined HOI classes. All three evaluation settings (*Setting 1–3*, introduced in Sec. 2.3) are constructed from the same set of 1,274 images to ensure comparability across settings.

## 2.3 BENCHMARK EVALUATION

**Evaluation Settings** Existing benchmarks face two major issues: missing interaction labels and unannotated human-object pairs. As discussed in Sec. 2.2, our negative-choice construction alleviates the first issue, but the second requires dedicated evaluation protocols. To address this, we design three evaluation settings that capture different aspects of HOI evaluation, all under the same multiple-choice format (see Fig. 2, with additional examples in Appendix C.2). **Setting 1** requires the model to predict interactions for all people in the image. This setting evaluates image-level HOI recognition while avoiding penalties for unannotated individuals through curated negative choices. **Setting 2** provides the bounding box of a target person, requiring the model to predict only that individual's interactions. This isolates individual interaction recognition from person detection and avoids errors from sparse annotations. **Setting 3** requires the model to first detect all persons and then predict the interactions for the one overlapping with the specified target person (IoU $\geq$ 0.5), thereby jointly evaluating detection and recognition. In both *Setting 2* and *Setting 3*, we adopt a human-centric formulation: only human bounding boxes are provided, and the model must implicitly infer the interacted object from its predicted interaction. This emphasizes reasoning about human interactions, while general object detection remains outside our scope and is already well studied with established benchmarks Deng et al. (2009b); Shao et al. (2019); Gupta et al. (2019b).

**Evaluation Metrics** To evaluate multiple-answer, multiple-choice questions, we adopt set-based metrics that directly compare the predicted and ground-truth label sets. These metrics are widely used in multi-label classification and question answering Rajpurkar et al. (2016); Wu & Zhou (2017). Specifically, we report: Instance-F1, Macro-F1, Micro-F1, and Exact Match Accuracy (EM).

Let $Q$ denote the set of all evaluation questions. For each question $q \in Q$, let $P_q$ be the set of predicted interaction labels and $G_q$ the ground-truth set of positive choices. Macro-F1 evaluates performance in a class-balanced manner. Let $\mathcal{C}$ denote the set of HOI classes. For each class $c \in \mathcal{C}$, we compute the F1-score over all questions involving $c$, denoted as $\text{F1}_c$. Macro-F1 is then obtained by averaging $\text{F1}_c$ across all classes so that rare and frequent HOI classes contribute equally:

$$\text{Macro-F1} = \frac{1}{|\mathcal{C}|} \sum_{c \in \mathcal{C}} \text{F1}_c = \frac{1}{|\mathcal{C}|} \sum_{c \in \mathcal{C}} \frac{2 \sum_q \mathbf{1}[c \in P_q \cap G_q]}{\sum_q \mathbf{1}[c \in P_q] + \sum_q \mathbf{1}[c \in G_q]}, \tag{1}$$

where $\mathbf{1}[\cdot]$ is the indicator function, which equals 1 if the condition is true and 0 otherwise.

Instance-F1 measures performance at the question level. For each $q$, we compute the F1-score between $P_q$ and $G_q$, and then average over all questions to obtain the overall score:

$$\text{Instance-F1} = \frac{1}{|Q|} \sum_{q \in Q} \text{F1}(q) = \frac{1}{|Q|} \sum_{q \in Q} \frac{2|P_q \cap G_q|}{|P_q| + |G_q|}. \quad (2)$$

Here $|\cdot|$ denotes set cardinality, and $P_q \cap G_q$ is the set of correctly predicted labels for question $q$.

Micro-F1 measures overall performance by aggregating predictions across all questions and computing a single F1-score from the total number of predicted and ground-truth labels:

$$\text{Micro-F1} = \frac{2 \sum_q |P_q \cap G_q|}{\sum_q |P_q| + \sum_q |G_q|}. \quad (3)$$

While F1 scores capture the balance between precision and recall at the global level, it does not indicate whether variations in performance are driven more by precision or recall. To provide complementary insights, we also report precision and recall averaged across all test questions, enabling separate examination of these two components.

**Exact Match Accuracy (EM).** Finally, we adopt Exact Match Accuracy (EM), which checks whether the predicted interaction set for a question exactly matches the ground-truth set. Unlike the exact-match mAP metric in traditional HOI benchmarks, which is often undermined by incomplete annotations and penalizes unlabeled interactions, our multiple-choice design mitigates this issue through curated negatives. Thus, EM provides a complementary measure of strict correctness: it reports how often the model's predictions are entirely correct.

$$\text{EM} = \frac{1}{|Q|} \sum_{q \in Q} \mathbf{1}[P_q = G_q]. \quad (4)$$

Together, these four metrics offer a comprehensive evaluation: Macro-F1 balances across classes, Instance-F1 captures per-question performance, Micro-F1 measures overall aggregate performance, and EM reflects exact prediction correctness.

## 3 EXPERIMENTS

### 3.1 EXPERIMENT SETUP

**Baselines** We evaluate two groups of baselines on our benchmark: general-purpose VLMs and HOI-specific methods. Recent large VLMs represent the frontier of general-purpose image understanding. Although not explicitly trained for HOI detection, they exhibit strong open-vocabulary grounding as well as visual and spatial reasoning abilities, making them natural candidates for HOI evaluation. Our VLM baselines include Qwen2/2.5-VL Bai et al. (2025), InternVL2.5/3 Chen et al. (2024); Wang et al. (2024b); Zhu et al. (2025), and LLaVA-OV Li et al. (2025). For HOI-specific methods, we include ADA-CM Lei et al. (2023), CMMP Lei et al. (2024c), LAIN Kim et al. (2025), HOLa Lei et al. (2025), and CMD-SE Lei et al. (2024b), which report competitive results on HICO-DET Chao et al. (2018) and SWiG-HOI Wang et al. (2022b). Further implementation details of all baselines are provided in Appendix D.1.

**Implementation Details** For general-purpose VLMs, we provide each question as a prompt together with explicit answer-format instructions. Model outputs are parsed accordingly and evaluated against the ground truth. For HOI-specific methods, we follow a top-$k$ matching strategy. Specifically, we take the top-5 predictions for each question and check whether any match the provided choices, consistent with the standard top-5 evaluation used in ImageNet Deng et al. (2009a). We adopt top-5 rather than top-1 to account for multiple correct answers per question, while top-10 would be unnecessarily permissive. Further implementation details are provided in Appendix D.1.

### 3.2 QUANTITATIVE RESULTS

**Finding 1: Difficulty in Disentangling Target Person Interactions for VLMs** Both small and large VLMs consistently struggle to separate the interactions of the target person from those of surrounding individuals. This limitation is evident when comparing *Setting 1* (Table 2) and *Setting 2*

| Method | Macro-F1 (%) | Instance-F1 (%) | Micro-F1 (%) | EM (%) | Avg. Prec. (%) | Avg. Rec. (%) |
|---|---|---|---|---|---|---|
| *HOI-specific methods* | | | | | | |
| ADA-CM | 45.89 | 56.23 | **67.49** | 11.85 | **83.16** | **56.78** |
| CMMP | 46.07 | 55.42 | 67.16 | 10.83 | 82.54 | 56.61 |
| LAIN | 44.27 | 53.87 | 65.06 | 10.52 | 80.61 | 54.54 |
| HOLa | 46.37 | 55.91 | 67.06 | 11.54 | 81.88 | **56.78** |
| CMD-SE | **46.51** | **57.25** | 66.49 | **14.13** | 83.05 | 55.44 |
| *VLM zero-shot evaluation* | | | | | | |
| InternVL2.5-38B | 51.96 | 51.81 | 55.31 | 17.27 | 84.47 | 41.11 |
| InternVL3-38B | 58.23 | 63.17 | 63.28 | 22.68 | **87.06** | 49.71 |
| Qwen2.5-VL-32B | **63.16** | **67.19** | **68.82** | **23.16** | 77.83 | **61.68** |
| LLaVA-OV-7B | 46.47 | 54.05 | 52.18 | 12.95 | **85.28** | 37.59 |
| InternVL3-8B | **55.52** | **61.17** | **61.57** | **20.41** | 83.51 | **48.76** |
| Qwen2-VL-7B | 41.92 | 36.91 | 41.64 | 7.14 | 82.75 | 27.82 |
| Qwen2.5-VL-7B | 51.29 | 56.04 | 56.03 | 14.84 | 80.15 | 43.06 |

Table 2: *Setting 1* experiment results comparison. Results are reported for VLMs and HOI-specific methods. Best performance within each group is highlighted in **bold**. "Avg. Prec." means the precision averaged across test set and "Avg. Rec." means the recall averaged across test set.

| Method | Macro-F1 (%) | Instance-F1 (%) | Micro-F1 (%) | EM (%) | Avg. Prec. (%) | Avg. Rec. (%) |
|---|---|---|---|---|---|---|
| *VLM zero-shot evaluation* | | | | | | |
| InternVL2.5-38B | 48.43 | 46.43 | 51.56 | 20.64 | 77.52 | 38.63 |
| InternVL3-38B | 58.94 | 67.41 | 67.81 | **35.64** | **81.90** | 57.85 |
| Qwen2.5-VL-32B | **62.90** | **69.52** | **70.69** | 35.01 | 75.30 | **66.61** |
| LLaVA-OV-7B | 47.76 | 56.53 | 54.80 | 25.12 | **77.43** | 42.40 |
| InternVL3-8B | **49.88** | 52.35 | 55.54 | 23.86 | 74.41 | 44.31 |
| Qwen2-VL-7B | 46.90 | 53.93 | 53.61 | 23.23 | 76.84 | 41.16 |
| Qwen2.5-VL-7B | 48.93 | **57.25** | **57.53** | **25.98** | 74.49 | **46.87** |

Table 3: *Setting 2* experiment results comparison. Best performance within each group is highlighted in **bold**. "Avg. Prec." means the precision averaged across test set and "Avg. Rec." means the recall averaged across test set.

(Table 3), which use the same images but differ in scope: *Setting 1* requires recognizing interactions for all people, whereas *Setting 2* isolates the target individual. Across models, we observe clear performance drops when moving from *Setting 1* to *Setting 2* (e.g., average precision decreases by 9.1% for InternVL3-8B, 5.7% for Qwen2.5-VL-7B, 5.2% for InternVL3-38B, and 2.5% for Qwen2.5-VL-32B). Error analysis confirms that 20–25% of mispredictions stem from attributing interactions of surrounding people to the target person (e.g., 21.6% for InternVL3-8B, 24.6% for Qwen2.5-VL-7B, 22.3% for InternVL2.5-38B, 24.2% for InternVL3-38B, and 22.0% for Qwen2.5-VL-32B). These results highlight a persistent weakness of VLMs in disentangling individual-level interactions in multi-person scenarios. Additional discussion and qualitative examples are provided in Appendix D.3.

**Finding 2: Detection Limitations of VLMs** The performance drop from *Setting 2* to *Setting 3* highlights that VLMs are far weaker at detection than at HOI recognition (e.g., Qwen2.5-VL-32B drops 16.6% Instance-F1). When ground-truth boxes are provided in *Setting 2*, VLMs can recognize interactions effectively, but performance drops significantly in *Setting 3*, where detection must be performed by the model itself (Table 4). HOI-specific methods, in contrast, integrate detection within their pipelines and therefore cannot directly exploit ground-truth boxes, but under *Setting 3* they often perform competitively, sometimes surpassing even large VLMs (e.g., ADA-CM achieves 13.6% higher Micro-F1 than InternVL3-38B and 0.3% higher than Qwen2.5-VL-32B). This indicates that detection remains a major bottleneck for VLMs, while HOI-specific models strike a more balanced trade-off between detection and recognition. Error analysis further shows VLMs particu-

| Method | Macro-F1 (%) | Instance-F1 (%) | Micro-F1 (%) | EM (%) | Avg. Prec. (%) | Avg. Rec. (%) |
|---|---|---|---|---|---|---|
| *HOI-specific methods* | | | | | | |
| ADA-CM | 43.02 | **47.76** | **61.69** | 19.15 | 76.25 | 51.80 |
| CMMP | 43.06 | 46.62 | 60.85 | 18.84 | 75.06 | 51.16 |
| LAIN | 41.28 | 45.64 | 59.09 | 19.31 | 73.42 | 49.44 |
| HOLa | 43.61 | 47.12 | 61.29 | 19.78 | 74.31 | **52.15** |
| CMD-SE | **47.49** | 44.66 | 58.71 | **20.33** | **78.33** | 46.96 |
| *VLM zero-shot evaluation* | | | | | | |
| InternVL2.5-38B | 43.31 | 19.97 | 29.05 | 9.18 | 85.18 | 17.51 |
| InternVL3-38B | 50.98 | 38.68 | 48.08 | 20.33 | **84.72** | 33.56 |
| Qwen2.5-VL-32B | **57.25** | **52.94** | **61.41** | **26.06** | 75.03 | **51.97** |
| LLaVA-OV-7B | - | - | - | - | - | - |
| InternVL3-8B | 31.89 | 4.96 | 8.51 | 2.04 | **76.09** | 4.51 |
| Qwen2-VL-7B | - | - | - | - | - | - |
| Qwen2.5-VL-7B | **42.96** | **30.53** | **37.49** | **14.29** | 75.92 | **24.89** |

Table 4: *Setting 3* experiment results comparison. Results are grouped by VLM and HOI-specific methods. Best performance within each group is highlighted in **bold**. "Avg. Prec." means the precision averaged across test set and "Avg. Rec." means the recall averaged across test set.

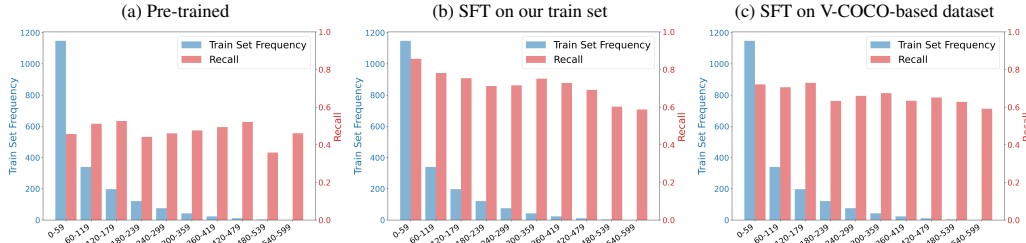

Figure 3: Comparison among pre-trained Qwen2.5-VL-7B Bai et al. (2025), SFT finetuned Qwen2.5-VL-7B on our train set and SFT finetuned Qwen2.5-VL-7B on V-COCO Lin et al. (2014). The blue histograms indicate the binned class frequency in our training dataset, while the red histograms present the recall rate. Head and tail classes are defined by HOI class frequency in our training set, and all HOI classes are ordered accordingly.

larly struggle in multi-person scenes, where they often miss or incompletely detect individuals, and in highly occluded cases. Qualitative examples are provided in Appendix D.3.

Tables 2, 3, and 4 together provide a comprehensive view of baseline performance. For HOI-specific methods (Tables 2 and 4), their Instance-F1 and Micro-F1 scores are comparable to those of small VLMs, but their Macro-F1 remains low, indicating poor balance across classes. Against large VLMs, they perform worse overall. For VLMs, performance shows a clear downward trend when moving from recognition-only (*Setting 1*) to specified-individual prediction (*Setting 2*) and further to detection-based evaluation (*Setting 3*). This progression demonstrates the strong recognition ability of large VLMs, while revealing their weakness once accurate localization or detection is required. Detailed quantitative discussions are included in Appendix D.2.

**Small Training Set Fine-tuning** We construct a training set from HICO-DET, yielding 111,459 training questions in total. This scale is modest compared to the massive datasets used in the pretraining of generative VLMs (details in Appendix C.3). When fine-tuned on our training set (Fig. 3(b)), recall (red bins) decreases steadily from head to tail classes, revealing a clear head-class bias. Fine-tuning on a different dataset (V-COCO), which follows a different distribution, produces a milder head-class bias than training on our set (Fig. 3(c)). In contrast, the VLM without fine-tuning shows no obvious head-class bias, as recall remains relatively flat across classes (Fig. 3(a)). Overall, while class imbalance has long been recognized as a challenge for HOI-specific methods, our benchmark

demonstrates that fine-tuned VLMs also inherit this limitation. Additional experimental results are provided in Appendix D.2.

## 4 RELATED WORK

**HOI Detection Methods** HOI detection methods are commonly divided into two-stage and one-stage approaches. Two-stage methods first detect humans and objects, then classify interactions between paired boxes Zhang et al. (2021; 2022); Park et al. (2023); Hou et al. (2022). One-stage methods instead predict ⟨human, object, verb⟩ triplets directly in an end-to-end manner Zou et al. (2021); Qu et al. (2022); Tu et al. (2023); Li et al. (2024b). Despite these advances, existing benchmarks still depend on exact matches with annotated HOIs, implicitly assuming exhaustive labeling of all human–object pairs. In practice, annotations are often incomplete, particularly in multi-person scenes. As a result, plausible but unlabeled interactions are penalized, leading to underestimated performance (e.g., below 50% mAP on HICO-DET for SOTA HOI methods). Our benchmark addresses this limitation by reformulating evaluation as a multiple-choice task with curated negatives, reducing the penalty on valid but unannotated predictions.

**Existing HOI Benchmarks** HICO-DET provides HOI annotations across 600 classes (117 verbs and 80 objects) Chao et al. (2018), with evaluation reported as mAP over all classes. V-COCO follows a similar protocol with 29 HOI classes defined on COCO images Lin et al. (2014). More recently, SWiG-HOI expands this setup to over 5,500 HOI classes for open-vocabulary evaluation Wang et al. (2022b). Although these benchmarks differ in label space, they all adopt exact-match evaluation, which requires predictions to align strictly with annotated HOI classes. This design is problematic: annotations are incomplete in temporally ambiguous scenarios and sparse in multi-person images, leaving many valid interactions unlabeled. Consequently, correct predictions are often penalized.

**Vision–Language Models for HOI** Large VLMs have recently become the frontier of general-purpose image understanding Bai et al. (2025); Zhu et al. (2025); Liu et al. (2024); Shahriar et al. (2024); OpenAI (2023); Yang et al. (2023). Although not explicitly trained for HOI detection, they exhibit strong open-vocabulary grounding as well as spatial and visual reasoning abilities, making them natural candidates for HOI understanding. However, their performance has not been systematically evaluated in HOI detection. Directly applying traditional benchmarks such as HICO-DET or V-COCO with exact-match mAP is unsuitable: VLMs can generate multiple valid interpretations beyond incomplete labels, and sparse annotations in multi-person scenes leave many pairs unlabeled. Both factors lead to underestimated performance. Our benchmark addresses these challenges by reformulating HOI detection as a multiple-choice task with curated negatives, and by introducing complementary evaluation settings that capture different aspects of HOI understanding.

## 5 CONCLUSION

In this work, we revisit the evaluation of HOI detection in the era of large VLMs. We showed that existing benchmarks such as HICO-DET suffer from incomplete annotations in two perspectives: missing interaction labels and sparse annotations in multi-person scenes, both of which lead to underestimated performance, particularly for VLMs. To address these issues, we introduced a benchmark that reformulates HOI detection as a multiple-answer, multiple-choice task with carefully curated negatives, ensuring that valid but unlabeled interactions are not penalized. Furthermore, we designed three complementary evaluation settings to assess different aspects of HOI tasks while mitigating the impact of sparse annotations. Our experiments demonstrate that large VLMs already surpass state-of-the-art HOI-specific methods on most metrics, even in zero-shot evaluation. Small VLMs are only on par with HOI-specific models and exhibit noticeable drops once detection ability is required. At the same time, VLMs show consistent shortcomings: misattributing the interactions of surrounding people to the target person, and weaker detection performance in complex multi-person or occluded cases. Overall, this benchmark establishes a foundation for advancing HOI detection in the era of VLMs and provides a unified protocol that encourages progress in both specialized HOI methods and general-purpose vision-language models.

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

| Time-sensitive HOI Classes | | | |
| --- | --- | --- | --- |
| boarding an airplane | breaking a baseball bat | catching a frisbee | catching a sports ball |
| controlling a tv | controlling a mouse | directing an airplane | directing a bus |
| dragging a suitcase | dragging a surfboard | dribbling a sports ball | exiting an airplane |
| flipping a skateboard | hitting a sports ball | hopping on a bicycle | hopping on a horse |
| hopping on a motorcycle | hopping on an elephant | kicking a sports ball | launching a boat |
| launching a kite | lifting a fork | losing an umbrella | moving a refrigerator |
| opening a backpack | opening a book | opening a bottle | opening a fire hydrant |
| opening a laptop | opening a microwave | opening a refrigerator | opening a scissors |
| opening a suitcase | opening a toilet | opening an oven | opening an umbrella |
| packing a suitcase | picking a banana | picking an apple | picking an orange |
| picking up a cake | picking up a donut | picking up a pizza | picking up a skateboard |
| picking up a skis | picking up a sports ball | picking up a suitcase | pulling a kite |
| pulling a tie | releasing a bird | serving a sports ball | setting a clock |
| setting an umbrella | sliding a pizza | squeezing an orange | swinging a baseball bat |
| swinging a tennis racket | swinging a remote | throwing a baseball bat | throwing a frisbee |
| throwing a sports ball | turning a motorcycle | tying a boat | tying a tie |
| waving a bus | wielding a baseball bat | wielding a knife | zipping a suitcase |

Table 5: List of 68 HOI classes in HICO-DET that are closely related to time information and often ambiguous in static images, while videos can disambiguate the temporal process.

## A   TIME-SENSITIVE HOI CLASSES IN HICO-DET

Out of the 600 pre-defined HOI classes in HICO-DET, we identified 68 classes that are closely related to time information, where static images can be inherently ambiguous and may not fully capture the temporal dynamics of the interaction. This finding shows that the examples highlighted in our paper (e.g., a frisbee interaction that could be interpreted as either "throwing" or "catching.") are not isolated incidents, but rather representative of a broader issue that affects a substantial portion of the benchmark. All these classes are listed in Table 5.

## B   HOI BENCHMARKING DATASET COMPARISON

Table 1 provides a systematic comparison of our benchmark with existing HOI datasets across multiple dimensions. In HICO-DET Chao et al. (2018), the majority of cases (60.2%) involve a single person interacting with a single object, which often results in relatively easy recognition. Our benchmark reduces this proportion to 33.3%, thereby shifting the focus toward more diverse and challenging scenarios. A key strength of our benchmark is the inclusion of multi-person images with different interactions. While only 7.5% of HICO-DET and 22.5% of V-COCO Lin et al. (2014) fall into this category, our dataset raises this to 38.9%, providing a richer evaluation of compositional reasoning across individuals. SWiG-HOI Wang et al. (2022b) and Bongard-HOI Jiang et al. (2022) contain no such cases, because they only provide one annotated person for each image, further distinguishing our benchmark.

Moreover, similar to existing HOI datasets, our dataset can be applied to HOI-specific methods, which ensures backward compatibility and allows direct comparison with existing HOI approaches. However, unlike HICO-DET, V-COCO, and SWiG-HOI, our dataset is explicitly designed to support general-purpose VLMs. This is achieved by framing the task as multiple-choice question answering, naturally aligning with the input–output format of modern VLMs. Bongard-HOI, although relevant for HOI recognition, is limited to binary classification tasks (i.e., "is this interaction present or not?"). Our dataset instead requires multi-class, multi-label prediction, reflecting the true complexity of HOI understanding in realistic images.

Taken together, our benchmarking dataset uniquely combines the strengths of previous HOI datasets while addressing their shortcomings. First, our datasets reduces over-simplified single-person cases, emphasizes multi-person cases with different interactions, remains compatible with HOI-specific methods, introduces explicit support for VLM evaluation, and requires full multi-class HOI pre-

diction. This makes it the first benchmark to emphasize challenging cases and enable comparison across both specialized HOI models and general-purpose VLMs.

## C  DATASET DETAILS

### C.1  DATASET LICENSES AND RELEASE

**Licenses** We use the HICO-DET dataset Chao et al. (2018), which is publicly released under a CC0: Public Domain license.

**Data Release and Ethical Considerations** We do not release data beyond the original HICO-DET dataset. Instead, our benchmark release will contain evaluation questions constructed on top of HICO-DET. Each question is associated with the corresponding image index in HICO-DET, and the ground-truth HOI annotations used in our evaluation. No images or raw annotations are redistributed and users are required to obtain HICO-DET separately under its original license. As our release only provides derived question–answer pairs and image index mappings, the risk of exposing personally identifiable information or offensive content is minimal. Consent considerations follow those of the original HICO-DET release, and we do not conduct an independent investigation of consent beyond the original dataset.

### C.2  DATASET EXAMPLES

Examples in Fig. 4 illustrate the main challenges our benchmark emphasizes. In multi-person scenarios (e.g., the bus, frisbee, and cell phone related examples), different individuals perform distinct interactions, which can confuse VLMs and lead to misattributing actions across people. At the same time, certain single-person cases are difficult due to either contactless interactions (e.g., peel apple) or visually similar categories (e.g., hold person vs. hug person). As a result, our benchmark creates a demanding evaluation of HOI understanding.

### C.3  OUR HOI TRAINING DATASET

We provide a training dataset with images selected from the HICO-DET training set. Since a model is expected to address all three settings, the training dataset combines the three types of questions: (i) a *Setting 1* question covering interactions of all annotated people in the image, (ii) a *Setting 2* question focusing on the interaction of the target person, and (iii) an additional detection question for *Setting 3*. This results in 111,459 training questions overall, which is relatively small comparedd to the billions of samples used in the pretraining of generative VLMs.

**Inherent Long-tailed Training Data in HOI** Since we observed the class imbalance problem after fine-tuning on HOI specific datasets, as shown in Fig. 3, a natural concern is why we do not re-balance our training set. Our data is constructed from HICO-DET, which is inherently long-tailed and serves as the standard benchmark in HOI detection. This imbalance is hard to remove because rare HOI classes are genuinely difficult to collect in the real world. Supporting this point, even when fine-tuned on V-COCO-based data with different HOI class definitions, models still show stronger performance on the head classes in our training set, shown in Fig. 3. These results indicate that long-tailed distributions are not unique to our training data but rather a common phenomenon that naturally arises in HOI due to the scarcity of rare interactions Yang et al. (2024); Li et al. (2022b); Hou et al. (2021).

## D  EXPERIMENTS

### D.1  IMPLEMENTATION DETAILS

**Baseline Details** We evaluate two groups of baselines on our benchmark: general-purpose VLMs and HOI-specific methods. Recent large VLMs represent the frontier of general-purpose image understanding. Qwen2-VL and Qwen2.5-VL (7B / 32B) Bai et al. (2025) are selected as they excel in fine-grained spatial localization and visual reasoning, making them suitable for HOI tasks. InternVL2.5 and InternVL3 (8B / 38B) Chen et al. (2024); Wang et al. (2024b); Zhu et al. (2025)

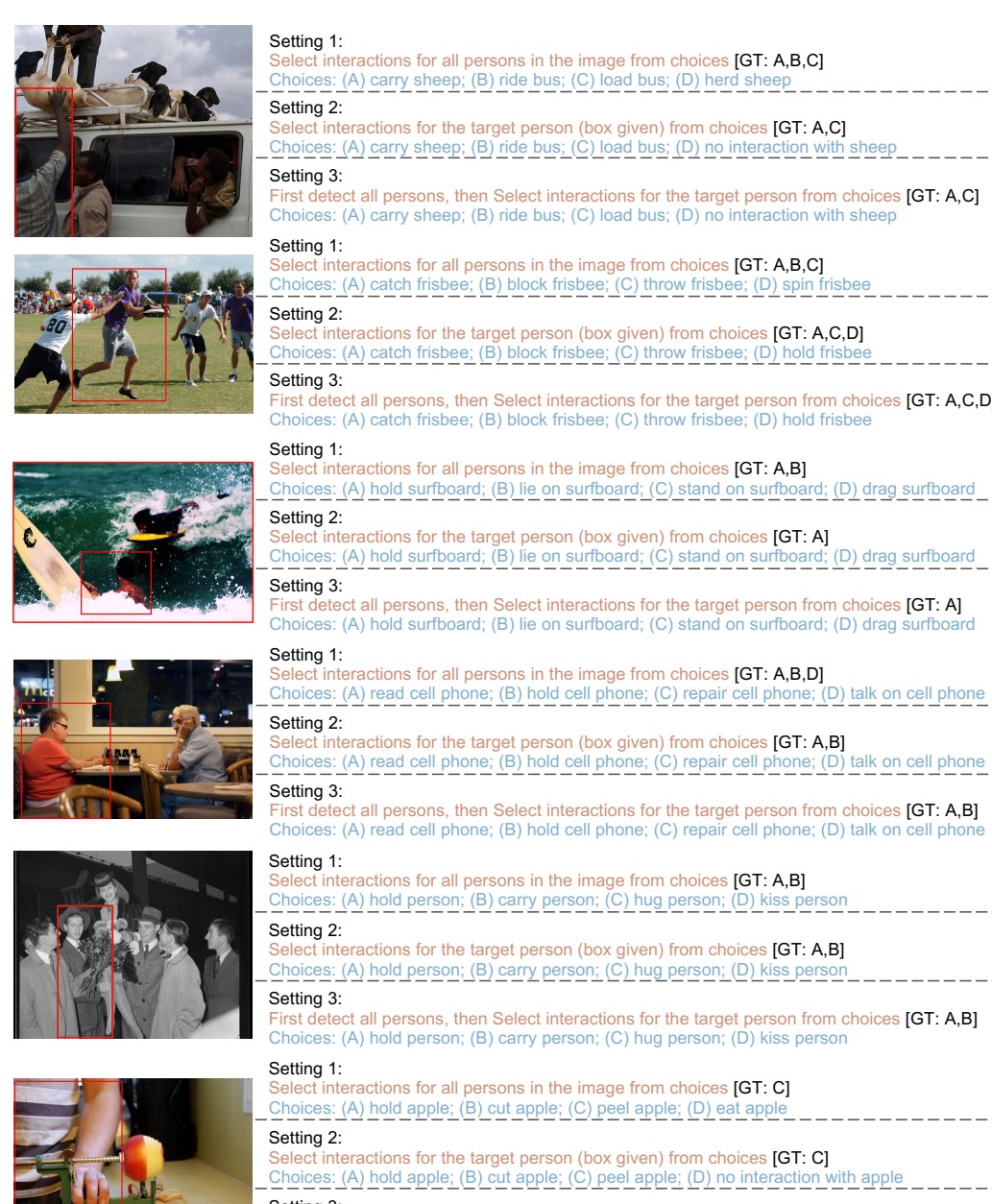

Figure 4: Example questions in our benchmark under the three evaluation settings.

are included because they achieve leading performance across diverse multimodal benchmarks and emphasize high-resolution perception, which is relevant for recognizing human–object interactions. LLaVA-OV-7B Li et al. (2025) is an instruction-tuned VLM designed for open-vocabulary understanding demonstrating versatility and strong performance across multiple vision–language tasks, making it a relevant baseline for HOI evaluation.

Beyond VLM baselines, we also evaluate recent HOI detection methods. ADA-CM Lei et al. (2023), CMMP Lei et al. (2024c), LAIN Kim et al. (2025) and HOLa Lei et al. (2025) demonstrate competitive performance on the existing HICO-DET benchmark Chao et al. (2018). In addition, CMD-SE Lei et al. (2024b) is a recent open-vocabulary HOI detection method emphasizing generalization ability, achieving competitive performance on SWiG-HOI Wang et al. (2022b) and HICO-DET

benchmarks. We use best-performing pre-trained checkpoints when available, and otherwise reproduce results with the authors' code under the closest available configurations. Specifically, ADA-CM, CMMP, and HOLa are evaluated with the ViT-L vision backbone, while CMD-SE and LAIN are based on ViT-B.

**Prompts for VLMs** For general-purpose VLMs, we provide the question prompt together with explicit answer-format instructions.

In *Setting 1*, the prompt is

*Question: Which of the following properly describes the interactions in the image ⟨image⟩? Choices: (A) ...,(B) ...,(C) ...,(D)... IMPORTANT: Reply with the letter(s) ONLY, separated by commas if multiple (e.g. A,B). For example, if correct answers are (A) and (B), your output must be: A,B Do NOT include any brackets or other symbols.*

In *Setting 2*, the prompt is

*Context: You are given an image ⟨image⟩ and a target person with a bounding box ⟨ human box ⟩. Question: Which of the following describes the interactions between the target person and any object in the image? Choices: (A) ...,(B) ...,(C) ...,(D)... IMPORTANT: Reply with the letter(s) ONLY, separated by commas if multiple (e.g. A,B). For example, if correct answers are (A) and (B), your output must be: A,B Do NOT include any brackets or other symbols.*

In *Setting 3*, we first obtain detection results with the following prompt:

*Provide the bounding box coordinates for every single person in the input image. The box coordinates represent as [x1, y1, x2, y2], where x is the horizontal pixel coordinate from the left edge, and y is the vertical pixel coordinate from the top edge. Return the detection results in JSON format strictly. For example: { "boxes": [[32, 109, 644, 418], [517, 0, 644, 23], [100, 50, 160, 200]]}.*

The subsequent question in *Setting 3* is identical to that in *Setting 2*, except that the ground-truth bounding box is replaced by the detected one. Therefore, we do not repeat the prompt here.

**Bounding Box Process for VLMs** In *Setting 2* and *Setting 3*, the input requires bounding boxes of the target person. Since different VLMs preprocess images in different ways, we adapt the bounding boxes accordingly to ensure consistency with the model input. Specifically, Qwen2/2.5-VL resizes input images such that both height and width are multiples of 28. We therefore resize the bounding boxes proportionally to the resized image coordinates. InternVL2.5/3 does not fix the image size but internally normalizes it. To align with its view of the image, we first query the model with a prompt asking for the perceived input resolution: *"Please provide the coordinates for the bottom-right point of the input image. Assume the coordinate system origin is at the top-left of the image, with x increasing to the right and y increasing downward. Return the coordinates as [width, height] in JSON format strictly. For example: [638, 415]."* Based on its returned size, we then rescale the bounding boxes into that coordinate system. LLaVA-OV takes the original image size directly as input. In this case, we use the original bounding boxes without additional processing. This preprocessing ensures that the bounding boxes we provide are always aligned with how each model internally processes the input image.

## D.2 ADDITIONAL QUANTITATIVE RESULTS AND DISCUSSION

In the following, we provide a detailed analysis of the results in Tables 2, 3, and 4.

In Table 2, *Setting 1* evaluates image-level HOI recognition. HOI-specific methods achieve competitive Instance-F1 and Micro-F1 compared to small VLMs, but their Macro-F1 is noticeably lower, reflecting limited class balance since Macro-F1 weights each HOI class equally regardless of frequency. Large VLMs (e.g., Qwen2.5-VL-32B, InternVL3-38B) outperform HOI-specific methods across all metrics, demonstrating strong generalization ability even without task-specific training, while smaller VLMs remain less robust.

In Table 3, *Setting 2* evaluates interaction prediction for a specified individual using the ground-truth bounding box. Large VLMs achieve notably higher EM accuracy (e.g., +12.96% for InternVL3-38B) compared to *Setting 1*, suggesting that predicting all interactions in *Setting 1* remains challenging. At the same time, other metrics decline, as models tend to predict more interactions from surrounding people, instead of focusing on the target person interactions. Nevertheless, large VLMs

| Method | Macro-F1 (%) | Instance-F1 (%) | Micro-F1 (%) | EM (%) | Avg. Prec. (%) | Avg. Rec. (%) |
|---|---|---|---|---|---|---|
| *VLM zero-shot evaluation* | | | | | | |
| InternVL2.5-38B | 48.12 | 46.69 | 52.26 | 21.04 | 78.25 | 39.23 |
| InternVL3-38B | 59.33 | 66.13 | 67.38 | **33.20** | **82.20** | 57.08 |
| Qwen2.5-VL-32B | **61.59** | **67.29** | **69.48** | 31.79 | 74.65 | **64.98** |
| LLaVA-OV-7B | 47.03 | 55.55 | 54.68 | 24.18 | **78.43** | 41.97 |
| InternVL3-8B | **51.81** | 53.47 | 57.05 | 24.25 | 76.77 | 45.41 |
| Qwen2-VL-7B | 46.06 | 51.85 | 52.71 | 21.51 | 77.28 | 40.00 |
| Qwen2.5-VL-7B | 49.56 | **56.91** | **57.98** | **25.35** | 75.22 | **47.17** |

Table 6: *Setting 3* experiment results comparison with off-the-shelf detector (DETR) Carion et al. (2020). Results are grouped by VLM size (small vs. large). Best performance within each group is highlighted in **bold**. "Avg. Prec." means the precision averaged across test set and "Avg. Rec." means the recall averaged across test set.

| Train set | Setting | Macro-F1 (%) | Instance-F1 (%) | Micro-F1 (%) | EM (%) | Avg. Prec. (%) | Avg. Rec. (%) |
|---|---|---|---|---|---|---|---|
| Ours | 1 | 69.98 | 75.14 | 78.28 | 32.26 | 77.27 | 79.32 |
| V-COCO-based | 1 | 64.00 | 68.74 | 70.85 | 22.61 | 75.98 | 66.38 |
| Ours | 2 | 63.89 | 68.66 | 72.27 | 26.69 | 66.11 | 79.70 |
| V-COCO-based | 2 | 60.65 | 64.77 | 67.46 | 26.92 | 68.43 | 66.52 |
| Ours | 3 | 62.63 | 59.82 | 69.65 | 24.88 | 68.48 | 70.86 |
| V-COCO-based | 3 | 58.21 | 54.87 | 63.54 | 22.53 | 70.03 | 58.15 |

Table 7: Experiment results comparison when Qwen2.5-VL-7B is fine-tuned on HOI datasets Carion et al. (2020) using Supervised Fine-Tuning (SFT). "Avg. Prec." means the precision averaged across test set and "Avg. Rec." means the recall averaged across test set.

consistently outperform smaller ones. HOI-specific methods are not directly comparable in this setting, since they cannot take ground-truth bounding boxes as input and typically rely on detector intermediate features.

As shown in Table 4, once detection is required in *Setting 3*, VLMs face notable drops in Instance-F1, Micro-F1 and EM compared to *Setting 2*, reflecting their inferior detection capability. HOI-specific methods, although less strong in the previous setting, become competitive here, with some even surpassing large VLMs in Micro-F1 and average recall. This indicates that detection remains a bottleneck for VLMs, while HOI-specific pipelines retain a more balanced trade-off between detection and recognition.

Since VLMs often struggle with reliable person detection, we follow the two-stage HOI detection paradigm Lei et al. (2023); Wang et al. (2024a); Zhang et al. (2022) and leverage a widely used off-the-shelf object detector, a DETR model Carion et al. (2020) pre-trained on HICO-DET. Table 6 shows that incorporating DETR helps improve performance in the *Setting 3* evaluation, though remains lower than in *Setting 2* due to detection errors. Among small models, Qwen2.5-VL-7B achieves the best overall performance among most evaluation metrics, except for Macro-F1 and average precision metrics. For large models, Qwen2.5-VL-32B outperforms InternVL3-38B most of the time, except for EM and average precision. By comparing Table 6 and Table 4, we observe a clear performance drop when VLMs perform detection on their own, as opposed to relying on off-the-shelf detectors (i.e., +6.6% Macro-F1 for Qwen2.5-Vl-7B, +4.34% for Qwen2.5-VL-32B, +19.92% for InternVL-8B and +8.35% for InternVL-38B). This highlights that the detection capability of current VLMs still lags behind that of specialized object detectors.

Table 7 compares Qwen2.5-VL-7B fine-tuned on our dataset versus a V-COCO-based dataset across the three evaluation settings. A clear trend emerges in the precision–recall balance. Before fine-tuning, VLMs typically exhibit much higher average precision than recall, reflecting a conservative prediction style that favors fewer outputs, shown in Table 2, 3 and 4. After training on our dataset, however, the model learns to adjust its strategy: recall surpasses precision across settings. This shift indicates that the model adapts to the multi-answer structure of our benchmark by predicting

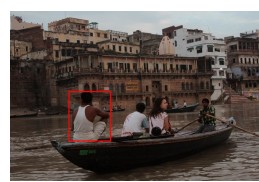
**Target person HOI:**
ride boat; sit on boat
**Other people HOI:**
row boat
**Predictions from VLM:**
sit on boat;
row boat (×)

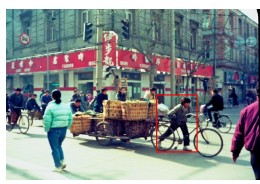
**Target person HOI:**
push bicycle
**Other people HOI:**
straddle bicycle
**Predictions from VLM:**
push bicycle;
straddle bicycle (×)

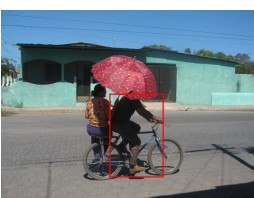
**Target person HOI:**
ride bicycle; hold bicycle
**Other people HOI:**
hold umbrella
**Predictions from VLM:**
ride bicycle;
hold umbrella (×)

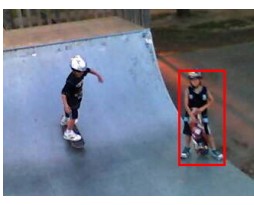
**Target person HOI:**
hold skateboard
**Other people HOI:**
ride skateboard
**Predictions from VLM:**
hold skateboard;
ride skateboard (×)

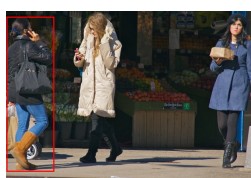
**Target person HOI:**
hold cell phone
**Other people HOI:**
text on cell phone
**Predictions from VLM:**
hold cell phone;
text on cell phone (×)

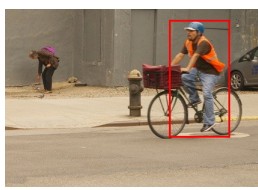
**Target person HOI:**
ride bicycle; sit on bicycle
**Other people HOI:**
wear backpack
**Predictions from VLM:**
ride bicycle;
wear backpack (×)

Figure 5: Illustration of failure cases of the Qwen2.5-VL-32B model in *Setting 2*, where they misattribute other people's interactions to the target person (in the red box).

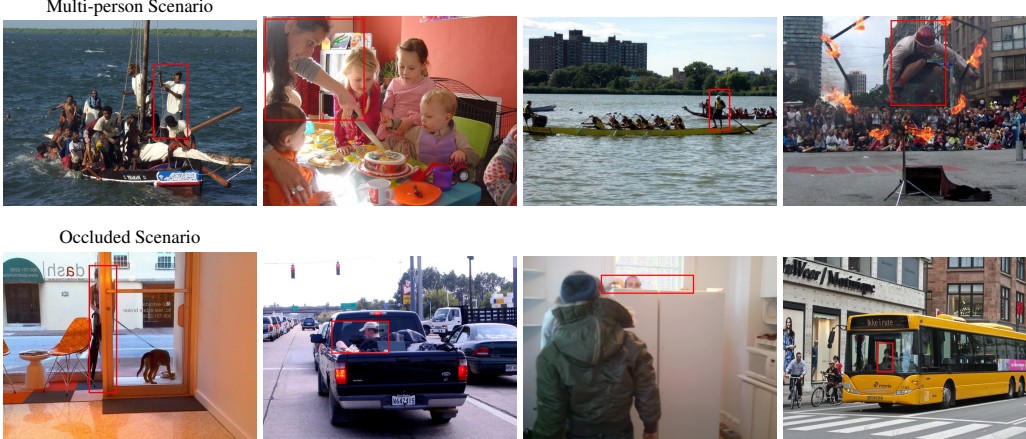

Figure 6: Failure detection cases of the Qwen2.5-VL-32B model in *Setting 3*. The red box marks the target person specified in the question. The first row shows failures in multi-person scenarios, while the second row shows failures under occlusion.

more for each question, which increases the chance of covering all correct answers. Although this comes at the cost of lower precision, it reflects an important bias learned from our dataset design, which emphasizes diverse and challenging cases where multiple valid answers frequently occur. In contrast, training on V-COCO-based data maintains a more conservative balance, but still acheive much higher recall than pre-trained VLMs.

### D.3 QUALITATIVE RESULTS

**Failure Cases on Distinguishing Target Person Interactions for VLM** Figure 5 provides concrete examples of the failure cases we discussed in the main text, where VLMs often misattribute interactions of surrounding people to the specified target person. In these cases, the target individual performs one action (e.g., ride boat or hold cell phone), while other people nearby perform different

actions (e.g., row boat or text on cell phone). The model incorrectly assigns these surrounding actions to the target person, leading to false positives. Such errors are more common when the target person shares the same object (e.g., bicycle, skateboard, cell phone) with others, or when nearby actions are visually salient but semantically irrelevant to the target. These examples confirm that individual interaction understanding in multi-person scenes pose a systematic challenge for VLMs, reinforcing our Finding 1 in the main paper.

**Failure Detection Cases for VLM** We use Qwen2.5-VL-32B model for the failure analysis, which achieves the highest performance in *Setting 3* among baselines in Table 4. As shown in Fig. 6, failures mainly occur in multi-person scenes, where the detector struggles to correctly localize individuals in crowded settings, and in occluded scenes, where heavy blockage of the person leads to missed or inaccurate bounding boxes. These cases show that complex layouts and occlusions remain key challenges for detection.

