# OpenReview forum: "Rethinking Human-Object Interaction Evaluation for both Vision-Language Models and HOI-Specific Methods"
_ICLR.cc/2026/Conference — ICLR 2026 Conference Withdrawn Submission_

### Official Review · Reviewer_94WA · 2025-10-28

**Soundness:** 3
**Presentation:** 2
**Contribution:** 2
**Rating:** 4
**Confidence:** 3

**Summary:**

This dataset work reformulates human-object interaction (HOI) detection as a multiple-answer, multiple-choice task to address issues of incomplete and ambiguous annotations in traditional benchmarks. Built upon HICO-DET, it introduces a more challenging test set by filtering simple cases, curating hard negatives, and providing three evaluation settings for a unified comparison of both vision-language models and specialized HOI methods. Results show that large VLMs already outperform HOI-specific methods on most metrics in zero-shot evaluation but struggle with disambiguating individual interactions in multi-person scenes and with detection in complex scenarios.

**Strengths:**

- The work compellingly addresses the critical issue of annotation limitations in existing HOI benchmarks.
- The proposed evaluation protocol is well-suited for assessing generative VLMs.

**Weaknesses:**

- Concerns about data construction. The benchmark construction pipeline relies heavily on a cascade of VLMs (GPT-4o, Qwen2.5-VL) to generate and filter negative choices. Despite the cross-checking mechanism, this design inherently risks error accumulation and introduces a circular dependency, as the quality of the benchmark is contingent on the judgment of the very models it seeks to evaluate.
- Potential Evaluation Bias Due to Benchmark Construction. Stemming from the above, a significant limitation is the lack of a human-verified clean subset to quantify the VLM-induced bias. Without a high-quality, expert-curated anchor set, it is impossible to disentangle how much of a model's high performance stems from genuine HOI understanding versus an overfitting to the specific biases and "blind spots" of the VLMs used in the benchmark's creation.
- Limited test set size. The filtered dataset only contains less than 25% original HICO-DET data, which raises a concern about whether this test set is enough to evaluate the  HOI ability.

**Questions:**

- Could the authors provide more analysis on the error accumulation issue in the data construction pipeline?
- Could the authors justify the potential evaluation bias?

---

### Official Review · Reviewer_NRdr · 2025-10-29

**Soundness:** 2
**Presentation:** 3
**Contribution:** 2
**Rating:** 2
**Confidence:** 5

**Summary:**

The paper identifies significant limitations in existing Human-Object Interaction (HOI) benchmarks, particularly concerning their inability to fairly evaluate modern Vision-Language Models (VLMs). The authors argue that issues like incomplete annotations and rigid exact-match evaluation protocols systematically underestimate the capabilities of generative models. To address this, they propose a new benchmark that reformulates HOI detection as a multiple-choice, multiple-answer task. This benchmark is constructed by curating challenging samples from HICO-DET and generating hard negatives using a VLM-based pipeline. Through experiments under three proposed evaluation settings, the authors conclude that large VLMs, even in a zero-shot setting, already surpass specialized HOI methods on this new benchmark, while also highlighting key VLM weaknesses in detection and multi-person interaction disentanglement.

**Strengths:**

1. The motivation behind this work is both timely and significant. The rapid advancement of VLMs necessitates a critical re-evaluation of long-standing computer vision tasks and their benchmarks. The paper correctly identifies a crucial disconnect between the generative, nuanced outputs of VLMs and the rigid, discrete evaluation criteria of traditional HOI datasets. The problem formulation, focusing on annotation sparsity and the penalization of valid but unannotated predictions, is a valuable contribution to the field.

2. The proposed three-setting evaluation protocol (image-level recognition, localized recognition, and detection-based recognition) is methodologically sound. It offers a structured way to disentangle the different capabilities of a model, providing more granular insights than a single end-to-end metric. The idea of reformulating the task as a multiple-choice question is a creative approach to handle the ambiguity and multi-label nature of HOI.

**Weaknesses:**

1. Unfair Comparison and Task Redefinition: The paper's central weakness lies in its evaluation framework, which seems fundamentally biased towards VLMs. By reformulating HOI detection as a multiple-choice classification problem, the benchmark largely abandons the detection aspect of the task, particularly the explicit localization of objects. Traditional HOI methods are designed to output <human_box, object_box, interaction_class> tuples. The proposed top-5 matching protocol is a post-hoc adaptation that forces these models into a paradigm they were not designed for, which is an inherently unfair comparison. This new framework effectively evaluates a different, simpler task—more akin to Visual Question Answering (VQA) conditioned on a person region—than the established HOI detection task.

2. Circular Reasoning in Benchmark Construction and Conclusions: The methodology for constructing the benchmark raises serious concerns about circular logic and self-fulfilling prophecy. The authors use a sophisticated pipeline of powerful VLMs (GPT-4.1, Qwen2.5-VL, GPT-4o) to curate the hard negative choices. The benchmark format is then redesigned to be optimally suited for VLM-style inputs and outputs. Consequently, the conclusion that VLMs perform better on this very benchmark is hardly surprising and lacks persuasive power. The experiment risks evaluating how well a given VLM aligns with the biases of the curator VLMs used to build the dataset, rather than its objective capability in HOI understanding.

3. Lack of Clarity and Inconsistent Narrative: The paper's organization and writing could be significantly improved. For example, the role of model fine-tuning is presented ambiguously. The main narrative champions the zero-shot capabilities of VLMs, but a the fine-tuning section is introduced late in the paper. Its purpose is not clearly integrated into the paper's main claims, making the overall message less coherent. Key implementation details, especially regarding the prompts and the complex multi-VLM pipeline for negative generation, could be described with greater clarity.

3. Insufficient Discussion of Relevant Literature: The paper fails to discuss or compare its approach with highly relevant recent works that also aim to leverage large language models for relational understanding and HOI detection like [1][2]. A thorough discussion of these alternative approaches is necessary to properly contextualize the paper's contribution and justify why a complete reformulation of the evaluation protocol is superior to methods that adapt language models to the existing HOI task.

4. Limited Justification and Scope: While the paper correctly identifies valid issues like temporal ambiguity in static images (e.g., throwing vs. catching), this justification feels insufficient to motivate the complete overhaul of the entire HOI evaluation framework. Furthermore, the new benchmark is constructed from a relatively small subset of 1,274 images from HICO-DET. Findings from such a small and heavily curated dataset may not be generalizable to the broader and more diverse challenges of in-the-wild HOI detection. The reliance on proprietary, large-scale models for data curation also poses a significant challenge for the benchmark's reproducibility and extensibility by the community.

[1] CoVLM: Composing Visual Entities and Relationships in Large Language Models Via Communicative Decoding. ICLR 2023.

[2] RelationLMM: Large Multimodal Model as Open and Versatile Visual Relationship Generalist. T-PAMI 2025.

**Questions:**

1. Could the authors provide a more robust justification for why the "top-5 matching" protocol is a fair and sufficient adaptation for traditional HOI detectors? Have they considered alternative ways to bridge the gap, or analyzed how this specific choice might disadvantage models designed for localization-centric, single-best-triplet prediction?

2. What is the authors' rationale for removing the explicit object localization requirement from the evaluation, a cornerstone of the traditional HOI detection task? If the goal is to evaluate interaction understanding, would it be more appropriate to frame this as a new task ("Human Interaction Recognition") rather than a new evaluation for an existing one?

3. How can the authors convince the reader that the benchmark's VLM-centric construction is not the primary reason for the conclusion that VLMs perform best? Are there any ablation studies or analyses that can demonstrate the objectivity of the curated negatives, independent of the VLM architecture used to generate them?

4. Could the authors elaborate on how their proposed framework and findings compare to recent works like RelationLMM, which also use LLMs for HOI but within a different paradigm? What are the relative advantages and disadvantages?

5. What is the primary takeaway the authors intend for readers to have from the fine-tuning experiment (Fig. 3)? Should this be interpreted as a limitation of fine-tuning for HOI, or a characteristic of the constructed training set? The connection to the main zero-shot evaluation thesis is unclear.

---

### Official Review · Reviewer_M8iM · 2025-10-31

**Soundness:** 2
**Presentation:** 3
**Contribution:** 2
**Rating:** 4
**Confidence:** 4

**Summary:**

This paper focuses on the human-object interaction detection problem, specifically benchmark evaluation related to this task. The authors first point out several limitations of the existing benchmark evaluations in the era of VLMs, then propose a multiple-choice evaluation benchmark for HOI and three evaluation settings to assess recognition, target-specific prediction, and joint detection. They evaluated several VLMs on the benchmark.

**Strengths:**

1. The paper is well-written and easy to understand.
2. The pointed out limitations of existing benchmarks are solid.

**Weaknesses:**

1. The constructed benchmark only include 1,274 images selected from the HICO-DET test set. As the number of evaluation samples is not quite large, and the images are from a single dataset, I'm concerned if it would capture the diverse human-object interactions in the real world and offer comprehensive evaluation.
2. Also, is this benchmark only for evaluation? As VLMs are not originally designed for HOI detection, they may need training samples for finetuning. The authors do perform small-scale fine-tuning experiments, but there is no standardized training split or guideline for future models to be adapted comparably. Providing an official training subset (or a clear protocol for generating one) would allow fair comparison between HOI-specific methods and VLMs after adaptation.
3. The authors mainly evaluated VLMs including Qwen2/2.5-VL, InternVL2.5/3, and LLaVA-OV. Some widely recognized VLMs such as Claude, GPT-4v are not evaluated.
4. Also, the author reported that VLMs fall short for detection. This is not surprising as VLMs are not designed to perform object detection. However, with the emergence of LLM-based agent systems, have the authors tried allowing VLMs to use tools such as detectors to perform HOI detection?
5. The authors mentioned in "Test Set Redistribution" that they "redistribute the test set to create a distribution that is less skewed and more distinct from the training set" of HICO-DET. This is quite confusing to me as in my opinion the evaluation set should reflect the real-world scenarios/distrituion, not to be less alike a specific training set.
6. The technical contribution of this paper seems limited as they only propose an evaluation benchmark curated from existing dataset.

**Questions:**

Please refer to the weakness. My major concern is the technical contribution.

---

### Official Review · Reviewer_SKFV · 2025-10-31

**Soundness:** 3
**Presentation:** 4
**Contribution:** 3
**Rating:** 2
**Confidence:** 5

**Summary:**

The authors' motivation is that current HOI dataset evaluations cannot fairly assess the performance of VLM-based expert HOI detectors and naive VLM models. They propose a new benchmark comprising three evaluation settings and balance the ratio of easy and difficult samples in the dataset. Finally, they evaluate popular VLM-based HOI detectors and VLMs, revealing the issues of VLMs in HOI detection.

**Strengths:**

•	The paper is presented very clearly, making it easy for readers to understand the motivation and solutions
•	The paper contains many exquisite charts and presents a large amount of data and experiments, with high credibility
•	The author identified a key issue in the field of HOI, which is how to evaluate the performance of traditional HOI detectors and current VLM models in the HOI field. They proposed a benchmark and balanced data distribution in the dataset. Their motivation is novel.

**Weaknesses:**

•	The current evaluation is still unfair. Firstly, the HOI field is divided into one-stage and two-stage methods. One-stage HOI detectors (e.g., UniHOI, BCHOI) heuristically discover relationships between humans and objects, while two-stage detectors (e.g., CLIP4HOI, CMMP) typically traverse all person-object pairs. This means locating humans and objects with interactions while ignoring those without, which is a major challenge in the field. However, in the paper, Setting1 is image-level, and Setting2 and Setting3 are given target bounding boxes while ignoring objects, which overlooks the fact that HOI is essentially a detection plus classification task.
•	The evaluation method proposed by the authors weakens the assessment of detection capability. Does this mean that it is essentially only evaluating the image-level classification ability of VLMs like CLIP, LLaVA, or Qwen? This evaluation result also aligns with the multimodal capabilities of these models.
•	In the process of evaluating the dataset, the authors assessed some traditional two-stage methods. I understand that this is done to better provide target bounding boxes, but one-stage VLM-based methods also need to be evaluated to make the benchmark comprehensive.
•	The implementation details described by the authors are vague. I understand that the VLM prompt provides multiple-choice options, but ordinary detectors only use top-5 recall results. The answers in the prompt provide hints to the VLM, which is very unfair.

**Questions:**

•	Discuss or explain how the evaluation strategy assesses the model's ability to discover human-object pairs with interactions, rather than merely classifying interactions, and how to truly establish a unified evaluation method for VLMs and traditional evaluators.
•	Explain the details of the evaluation clearly.

---

### Note · Authors · 2025-11-12

I have read and agree with the venue's withdrawal policy on behalf of myself and my co-authors.